# DISTRIBUTIONAL REINFORCEMENT LEARNING FOR RISK-SENSITIVE POLICIES

## ABSTRACT

We address the problem of learning a risk-sensitive policy based on the CVaR risk measure using distributional reinforcement learning. In particular, we show that applying the distributional Bellman optimality operator with respect to a risk-based action-selection strategy overestimates the dynamic, Markovian CVaR. The resulting policies can however still be overly conservative and one often prefers to learn an optimal policy based on the static, non-Markovian CVaR. To this end, we propose a modification to the existing algorithm and show that it can indeed learn a proper CVaR-optimized policy. Our proposed approach is a simple extension of standard distributional RL algorithms and can therefore take advantage of many of the recent advances in deep RL. On both synthetic and real data, we empirically show that our proposed algorithm is able to produce a family of risk-averse policies that achieves a better tradeoff between risk and the expected return.

## 1 INTRODUCTION

In standard reinforcement learning (RL) (Sutton & Barto, 2018), one seeks to learn a policy that maximizes an objective, usually the expected total discounted rewards or the long-term average rewards. In stochastic domains, especially when the level of uncertainty involved is high, maximizing the expectation may not be the most desirable since the solution may have high variance and occasionally performs badly. In such scenarios one may choose to learn a policy that is more risk-averse and avoids bad outcomes, even though the long-term average performance is slightly lower than the optimal.

In this work we consider optimizing the conditional value-at-risk (CVaR) (Rockafellar & Uryasev, 2000), a popular risk measure, widely used in financial applications, and is increasingly being used in RL. The CVaR objective focuses on the lower tail of the return and is therefore more sensitive to rare but catastrophic outcomes. Various settings and RL approaches have been proposed to solve this problem (Petrik & Subramanian, 2012; Chow & Ghavamzadeh, 2014; Chow & Pavone, 2014; Tamar et al., 2015; Tamar et al., 2017; Huang & Haskell, 2020). Most of the proposed approaches, however, involve more complicated algorithms than standard RL algorithms such as Q-learning (Watkins & Dayan, 1992) and its deep variants, e.g. DQN (Mnih et al., 2015).

Recently, the distributional approach to RL (Bellemare et al., 2017; Morimura et al., 2010) has received increased attention due to its ability to learn better policies than the standard approaches in many challenging tasks (Dabney et al., 2018a;b; Yang et al., 2019). Instead of learning a value function that provides the expected return of each state-action pair, the distributional approach learns the entire return distribution of each state-action pair. While this is computationally more costly, the approach itself is a simple extension to standard RL and is therefore easy to implement and able to leverage many of the advances in deep RL.

Since the entire distribution is available, one naturally considers exploiting this information to optimize for an objective other than the expectation. Dabney et al. (2018a) presented a simple way to do so for a family of risk measures including the CVaR. The theoretical properties of such approach, however, are not clear. In particular, it is not clear whether the algorithm converges to any particular variant of CVaR-optimal policy. We address this issue in this work.

Our main contribution is to first show that the proposed algorithm in (Dabney et al., 2018a) overestimates the dynamic, Markovian CVaR but empirically can be as conservative. It has been demon-

strated that this variant of CVaR can be overly conservative in many scenarios (Tamar et al., 2017; Yu et al., 2017), and one may prefer the static CVaR instead as the objective. Our second contribution is to propose a modified algorithm that can help achieve this. Empirically, we show that the proposed approach learns policies that perform better in terms of the overall CVaR objective on both synthetic and real-world problems.

We close the introduction section with some references to related works. We formally present our problem setup as well as our main analytical results in Section 2. Section 3 describes our proposed algorithm and finally, Section 4 presents our empirical results.

## 1.1 RELATED WORKS

The literature on distributional RL has been greatly expanded recently (Morimura et al., 2010; Bellemare et al., 2017; Barth-Maron et al., 2018; Dabney et al., 2018a;b; Yang et al., 2019). Most of these works focus on the modeling aspects, such as the choice of representations for the value distributions. The approach has been used to enhance exploration in RL (Mavrin et al., 2019; Zhang & Yao, 2019) and in risk-sensitive applications (Wang et al., 2019; Bernhard et al., 2019).

Solving Markov decision processes (MDP) with risk-sensitive objectives have been addressed in many works (Howard & Matheson, 1972; Ruszczynski, 2010; Bäuerle & Ott, 2011), including RL approaches (Borkar, 2001; Tamar et al., 2012; L.A. & Ghavamzadeh, 2013). In particular, Chow & Ghavamzadeh (2014); Tamar et al. (2015) deal with the static CVaR objectives while Petrik & Subramanian (2012); Chow & Pavone (2014) deal with the dynamic CVaR objectives. Tamar et al. (2017) proposed a policy-gradient approach that deals with both the static and the dynamic CVaR objectives. Closest to ours is the work by Stanko & Macek (2019). Their proposed approach also makes use of distributional RL but it is not clear whether their action selection strategy properly optimizes either the static or the dynamic CVaR.

## 2 PROBLEM SETUP AND MAIN RESULTS

We consider a discrete-time Markov decision process with state space $X$ and action space $A$. For simplicity we assume that $X$ and $A$ are finite, although our results and algorithm can be readily extended to more general state-action spaces. We assume that the rewards are bounded and drawn from a countable set $R \subset \mathbb{R}$. Given states $x_t, x_{t+1} \in X$ for any $t \in \{0, 1, \ldots\}$, the probability of receiving reward $r_t \in R$ and transitioning to $x_{t+1}$ after executing $a_t \in A$ in $x_t$ is given by $p(r_t, x_{t+1} | x_t, a_t)$. Without loss of generality we assume a fixed initial state $x_0$, unless stated otherwise. Given a policy $\pi : H \to \mathcal{P}(A)$, where $H$ is the set of all histories so far $h_t := (x_0, a_0, r_0, x_1, a_1, r_1, \ldots, x_t) \in H$, and $\mathcal{P}(A)$ the space of distributions over $A$, its expected total discounted reward over time is given by

$$V^\pi := \mathbb{E}_p^\pi \left[ \sum_{t=0}^\infty \gamma^t r_t \right]$$

where $\gamma \in (0, 1)$ is a discount factor. The superscript $\pi$ in the expectation indicates that the actions $a_t$ are drawn from $\pi(h_t)$. The subscript $p$ indicates that the rewards and state transitions are induced by $p$.

In standard RL, we aim to find a policy that maximizes $V^\pi$. It is well-known that there exists a deterministic stationary policy $\pi : X \to A$ whose decisions depend only on the current state, that gives optimal $V^\pi$, and therefore one typically works in the space of stationary deterministic policies. Key to a dynamic-programming solution to the above problem is the use of a value function $Q^\pi(x, a) := \mathbb{E}_p^\pi \left[ \sum_{t=0}^\infty \gamma^t r_t \mid x_0 = x, a_0 = a \right]$, which satisfies the Bellman equation

$$\forall x, a, \quad Q^\pi(x, a) = \sum_{r, x'} p(r, x' | x, a) \left[ r + \gamma Q^\pi(x', \pi(x')) \right]. \tag{1}$$

The optimal value $Q^*(x, a) := Q^{\pi^*}(x, a)$ for any optimal policy $\pi^*$ satisfies the Bellman optimality equation

$$\forall x, a, \quad Q^*(x, a) = \sum_{r, x'} p(r, x' | x, a) \left[ r + \gamma \max_{a'} Q^*(x', a') \right]. \tag{2}$$

Furthermore, for any $Q$-function $Q \in \mathcal{Q} := \{q : X \times A \to \mathbb{R} \mid q(x,a) < \infty, \forall x, a\}$, one can show that the operator $\mathcal{T}^\pi$ defined by $\mathcal{T}^\pi Q(x,a) := \sum_{r,x'} p(r, x'|x, a)[r + \gamma Q(x', \pi(x'))]$ is a $\gamma$-contraction in the sup-norm $\|Q\|_\infty := \max_{x,a} |Q(x,a)|$ with fixed-point satisfying (1). One can therefore start with an arbitrary $Q$-function and repeatedly apply $\mathcal{T}^\pi$, or its stochastic approximation, to learn $Q^\pi$. An analogous operator $\mathcal{T}$ can also be shown to be a $\gamma$-contraction with fixed-point satisfying (2).

## 2.1 STATIC AND DYNAMIC CVAR

The expected return $V^\pi$ is risk-neutral in the sense that it does not take into account the inherent variability of the return. In many application scenarios, one may prefer a policy that is more risk-averse, with better sensitivity to bad outcomes. In this work, we focus on the conditional value-at-risk (CVaR), which is a popular risk measure that satisfies the properties of being coherent (Artzner et al., 1999). The $\alpha$-level CVaR for a random real-valued variable $Z$, for $\alpha \in (0, 1]$, is given by (Rockafellar & Uryasev, 2000)

$$C_\alpha(Z) := \max_{s \in \mathbb{R}} \; s - \frac{1}{\alpha}\mathbb{E}[(s - Z)^+]$$

where $(x)^+ = \max\{x, 0\}$. Note that we are concerned with $Z$ that represents returns (the higher, the better), so this particular version of CVaR focuses on the lower tail of the distribution. In particular, the function $s \mapsto s - \frac{1}{\alpha}\mathbb{E}[(s - Z)^+]$ is concave in $s$ and the maximum is always attained at the $\alpha$-level quantile, defined as

$$q_\alpha(Z) := \inf\{s : \Pr(Z \leq s) \geq \alpha\}.$$

For $\alpha = 1$, $C_\alpha$ reduces to the standard expectation. In the case $Z$ is absolutely continuous, we have the intuitive $C_\alpha(Z) = \mathbb{E}[Z|Z < q_\alpha]$.

Our target random variable is the total discounted return $Z^\pi := \sum_{t=0}^\infty \gamma^t r_t$ of a policy $\pi$, and our objective is to find a policy that maximizes $C_\alpha(Z^\pi)$, where the optimal CVaR is given by

$$\max_\pi \max_s \; s - \frac{1}{\alpha}\mathbb{E}_p^\pi[(s - Z^\pi)^+]. \tag{3}$$

In the context where $Z$ is accumulated over multiple time steps, the objective (3) corresponds to maximizing the so-called static CVaR. This objective is time-inconsistent in the sense that the optimal policy may be history-dependent and therefore non-Markov. This is, however, perfectly expected since the optimal behavior in the later time steps may depend on how much rewards have been accumulated thus far – more risky actions can be taken if one has already collected sufficiently large total rewards, and vice versa. From the point of view of dynamic programming, an alternative, time-consistent or Markovian version of CVaR may be more convenient. A class of such risk measures was proposed by Ruszczynski (2010), and we shall refer to this version of CVaR as the dynamic CVaR, defined recursively as[1]

$$\forall \pi, x, a, \quad D_{\alpha,0}^\pi(x,a) := C_\alpha[r_t | x_t = x, a_t = a],$$

$$\forall \pi, x, a, T > 0, \quad D_{\alpha,T}^\pi(x,a) := C_\alpha[r_t + \gamma D_{\alpha,T-1}^\pi(x_{t+1}, \pi(x_{t+1}))|x_t = x, a_t = a], \quad \text{and}$$

$$\forall \pi, x, a, \quad D_\alpha^\pi(x,a) := \lim_{T \to \infty} D_{\alpha,T}^\pi(x,a).$$

It can be shown (Ruszczynski, 2010) that there exists a stationary deterministic optimal policy $\pi^*$, maximizing $D_\alpha^\pi(x,a)$ for all $x, a$, whose dynamic CVaR is given by $D_\alpha^* := D_\alpha^{\pi^*}$. In particular, the operator $\mathcal{T}_{D,\alpha}$ defined by

$$\mathcal{T}_{D,\alpha} D(x,a) := C_\alpha[r_t + \gamma \max_{a'} D(x_{t+1}, a')|x_t = x, a_t = a] \tag{4}$$

for $D \in \mathcal{Q}$ is a $\gamma$-contraction in sup-norm with fixed-point satisfying

$$\forall x, a, \quad D_\alpha^*(x,a) = C_\alpha[r_t + \gamma \max_{a'} D_\alpha^*(x_{t+1}, a')|x_t = x, a_t = a]. \tag{5}$$

---

[1] We use a slightly different definition from that in (Ruszczynski, 2010), but conceptually they are essentially the same.

The dynamic CVaR, however, can be overly conservative in many cases. We illustrate this with some empirical results in Section 4. In such cases it may be favorable to use the static CVaR. Bäuerle & Ott (2011) suggest an iterative process that can be used to solve for the optimal static CVaR policy. The approach is based on (3):

1. For a fixed $s$, one can solve for the optimal policy with respect to $\max_\pi \mathbb{E}[-(s - Z^\pi)^+]$.

2. For a fixed $\pi$, the optimal $s$ is given by the $\alpha$-level quantile of $Z^\pi$.

3. Repeat until convergence.

Step one above can be done by solving an augmented MDP with states $\tilde{x} = (x, s) \in X \times \mathbb{R}$, where $s$ is a moving threshold keeping track of the accumulated rewards so far. In particular, this MDP has no rewards and state transition is given by $p(0, (x', \frac{s-r}{\gamma})|(x, s), a) := p(r, x'|x, a)$. Solving this augmented MDP directly using RL, however, can result in poor sample efficiency since each example $(x, a, r, x')$ may need to be experienced many times under different threshold $s$. In this work, we propose an alternative solution using the approach of distributional RL.

## 2.2 DISTRIBUTIONAL RL

In standard RL, one typically learns the $Q^\pi(x, a)$ value for each $(x, a)$ through some form of temporal-difference learning (Sutton & Barto, 2018). In distributional RL (Bellemare et al., 2017), one instead tries to learn the entire distribution of possible future return $Z^\pi(x, a)$ for each $(x, a)$. The $Q$-value can then be extracted by simply taking the expectation $Q^\pi(x, a) = \mathbb{E}[Z^\pi(x, a)]$.

The objects of learning are distribution functions $U \in \mathcal{Z} := \{Z : X \times A \to \mathcal{P}(\mathbb{R}) \mid \mathbb{E}[|Z(x, a)|^q] < \infty, \forall x, a, q \geq 1\}$. For any state-action pair $(x, a)$, we use $U(x, a)$ to denote a random variable with the respective distribution. Let $\widetilde{\mathcal{T}}^\pi$ be the distributional Bellman operator on $\mathcal{Z}$ such that

$$\widetilde{\mathcal{T}}^\pi U(x, a) \overset{D}{:=} R + \gamma U(X', \pi(X'))$$

where $D$ denotes equality in distribution, generated by the random variables $R, X'$ induced by $p(r, x'|x, a)$. We use the notation $\widetilde{\mathcal{T}}$ instead of $\mathcal{T}$ when referring to a distributional operator, where $\widetilde{\mathcal{T}}^\pi U(x, a)$ is a random variable. (Bellemare et al., 2017) show that $\widetilde{\mathcal{T}}^\pi$ is a $\gamma$-contraction in $\mathcal{Z}$ in the following distance metric

$$d(U, V) := \sup_{x, a} W(U(x, a), V(x, a))$$

where $W$ is the 1-Wasserstein distance between the distributions of $U(x, a)$ and $V(x, a)$. Furthermore, the operator $\widetilde{\mathcal{T}}$ defined by

$$\widetilde{\mathcal{T}}U(x, a) \overset{D}{:=} R + \gamma U(X', A'), \quad A' = \arg\max_{a'} \mathbb{E}[U(X', a')] \tag{6}$$

can be shown to be a $\gamma$-contraction in $\mathcal{Q}$ in sup-norm under element-wise expectation, i.e.,

$$\|\mathbb{E}\widetilde{\mathcal{T}}U - \mathbb{E}\widetilde{\mathcal{T}}V\|_\infty \leq \gamma\|\mathbb{E}U - \mathbb{E}V\|_\infty,$$

where $\mathbb{E}\widetilde{\mathcal{T}}U \in \mathcal{Q}$ such that $\mathbb{E}\widetilde{\mathcal{T}}U(x, a) := \mathbb{E}[\widetilde{\mathcal{T}}U(x, a)]$, and $\mathbb{E}U, \mathbb{E}V, \mathbb{E}\widetilde{\mathcal{T}}V$ all similarly defined. In general, $\widetilde{\mathcal{T}}$ is not expected to be a contraction in the space of distributions for the obvious reason that multiple optimal policies can have very different distributions of the total return even though they all have the same expected total return.

Since one keeps the full distribution instead of just the expectation, a natural way to exploit this is to extract more than just the expectation from each distribution. In particular, in (6), one can select the action $a'$ based on measures other than the expectation $\mathbb{E}[U(x', a')]$. This is done by Dabney et al. (2018a) where a distortion measure on the expectation is used to select actions using various risk measures on $U(x', a')$, including the CVaR. If we replace $\mathbb{E}[U(x', a')]$ with $C_\alpha[U(x', a')]$, one may guess that it converges to the optimal dynamic CVaR policy satisfying (5). This is however, not true in general. We now show that choosing actions using $C_\alpha[U(x', a')]$ results in overestimating the dynamic CVaR value $D_\alpha^\pi$ and $D_\alpha^*$.

**Proposition 1.** *Let $U \in \mathcal{Z}$. Let $C_\alpha[U] \in \mathcal{Q}$ such that $C_\alpha[U](x,a) := C_\alpha[U(x,a)]$. Let $\mathcal{T}_{D,\alpha}$ as defined in (4). The distributional Bellman operator $\widetilde{\mathcal{T}}_{D,\alpha}$ given by*

$$\widetilde{\mathcal{T}}_{D,\alpha}U(x,a) \overset{D}{:=} R + \gamma U(X', A'), \quad A' = A(X') := \arg\max_{a'} C_\alpha[U(X', a')]$$

*satisfies*

$$\forall x, a, \quad C_\alpha[\widetilde{\mathcal{T}}_{D,\alpha}U(x,a)] \geq (\mathcal{T}_{D,\alpha}C_\alpha[U])(x,a).$$

*Similarly, for a fixed $\pi$, we have that*

$$\forall x, a, \quad C_\alpha[\widetilde{\mathcal{T}}^\pi_{D,\alpha}U(x,a)] \geq (\mathcal{T}^\pi_{D,\alpha}C_\alpha[U])(x,a).$$

*Proof.* We will use the properties of CVaR as a coherent risk measure (Artzner et al., 1999). In particular, $C_\alpha(Z)$ is concave in $Z$ (recall that we use the lower-tail version) where $\forall \lambda \in [0,1], C_\alpha(\lambda Z_1 + (1-\lambda)Z_2) \geq \lambda C_\alpha(Z_1) + (1-\lambda)C_\alpha(Z_2)$, and satisfies both translation invariance and positive homogeneity, where $\forall c, \lambda \in \mathbb{R}, \lambda \geq 0, C_\alpha(\lambda Z + c) = \lambda C_\alpha(Z) + c$.

$$
\begin{aligned}
C_\alpha[\widetilde{\mathcal{T}}_{D,\alpha}U(x,a)] = C_\alpha &\left[ \sum_{r,x'} p(r,x'|x,a)\left[r + \gamma U(x', A(x'))\right] \right] \\
&\overset{(a)}{\geq} \sum_{r,x'} p(r,x'|x,a) C_\alpha[r + \gamma U(x', A(x'))] \\
&\overset{(b)}{=} \sum_{r,x'} p(r,x'|x,a)\left[r + \gamma C_\alpha[U(x', A(x'))]\right] \\
&= \mathbb{E}\left[R + \gamma C_\alpha[U(X', A(X'))]\right] \\
&\overset{(c)}{\geq} C_\alpha\left[R + \gamma C_\alpha[U(X', A(X'))]\right] \\
&= (\mathcal{T}_{D,\alpha}C_\alpha[U])(x,a)
\end{aligned}
$$

where we use the coherent properties of $C_\alpha$ in $(a)$ and $(b)$, and in $(c)$ we use the fact that the expectation $\mathbb{E}$ is $C_\alpha$ for $\alpha = 1$ and upperbounds all other $C_\alpha$. The proof for the fixed $\pi$ case is analogous. $\qquad\square$

It is easy to construct an example where the inequalities in Proposition 1 are strict and that $\widetilde{\mathcal{T}}_{D,\alpha}$ converges to a policy that is different from the optimal $D^*_\alpha$ policy. Unfortunately, through empirical observations, $\widetilde{\mathcal{T}}_{D,\alpha}$ still results in policies that are closer to those optimizing the dynamic CVaR rather than the static CVaR.

It is now natural to ask whether we can optimize for the static CVaR instead while still staying within the framework of distributional RL. Recall that it is possible to optimize for the static CVaR by solving an augmented MDP as part of an iterative process. Instead of explicitly augmenting the state space, we rely on the distributions $U \in \mathcal{Z}$ to implicitly "store" the information needed. This approach will make the most of each transition example from $(x,a)$, since it updates an entire distribution, and indirectly the entire set of states $(x,s)$ for all $s$ in the augmented MDP. For this, the action selection strategy in (6) plays a critical role.

Given $U \in \mathcal{Z}$ and $s \in \mathbb{R}$, define $\zeta(U,s) \in \mathcal{Q}$ such that $\zeta(U,s)(x,a) := \mathbb{E}[-(s - U(x,a))^+]$. We define a distributional Bellman operator for the threshold $s$ as follows,

$$\widetilde{\mathcal{T}}_s U(x,a) \overset{D}{:=} R + \gamma U(X', A'), \quad A' = A^U_s(R, X') := \arg\max_{a'} \zeta\left(U, \frac{s-R}{\gamma}\right)(X', a').$$

The following result shows that at least for a fixed target threshold, improvement is guaranteed after each application of $\widetilde{\mathcal{T}}_s$.

**Proposition 2.** *For any $U, V \in \mathcal{Z}$, and any $s \in \mathbb{R}$,*

$$\left\|\zeta(\widetilde{\mathcal{T}}_s U, s) - \zeta(\widetilde{\mathcal{T}}_s V, s)\right\|_\infty \leq \gamma \sup_{s'} \left\|\zeta(U, s') - \zeta(V, s')\right\|_\infty.$$

*Proof.* For each $(x, a)$,

$$\left| \zeta(\widetilde{\mathcal{T}}_s U, s)(x, a) - \zeta(\widetilde{\mathcal{T}}_s V, s)(x, a) \right|$$

$$= \left| \sum_{r, x'} p(r, x' | x, a) \left( \int -(s - (r + \gamma u))^+ dU(x', A_s^U(r, x')) + \int (s - (r + \gamma v))^+ dV(x', A_s^V(r, x')) \right) \right|$$

$$= \gamma \left| \sum_{r, x'} p(r, x' | x, a) \left( \int - \left( \frac{s - r}{\gamma} - u \right)^+ dU(x', A_s^U(r, x')) + \int \left( \frac{s - r}{\gamma} - v \right)^+ dV(x', A_s^V(r, x')) \right) \right|$$

$$= \gamma \left| \sum_{r, x'} p(r, x' | x, a) \left( \max_{a'} \zeta \left( U, \frac{s - r}{\gamma} \right) (x', a') - \max_{a''} \zeta \left( V, \frac{s - r}{\gamma} \right) (x', a'') \right) \right|$$

$$\overset{(a)}{\leq} \gamma \sum_{r, x'} p(r, x' | x, a) \max_{a'} \left| \zeta \left( U, \frac{s - r}{\gamma} \right) (x', a') - \zeta \left( V, \frac{s - r}{\gamma} \right) (x', a') \right|$$

$$\leq \gamma \sup_{x', a', s'} | \zeta(U, s')(x', a') - \zeta(V, s')(x', a') |$$

where in $(a)$ we use triangle inequality and the fact that $| \max_a f(a) - \max_{a'} g(a') | \leq \max_a | f(a) - g(a) |$. □

Since we only keep one distribution for each $(x, a)$, we can only apply $\widetilde{\mathcal{T}}_s$ for a single chosen $s$ during each update. Applying $\widetilde{\mathcal{T}}_s$ can potentially change $\zeta(\widetilde{\mathcal{T}}_s U, s')(x, a)$ for any other $s'$ and there is no guarantee that similar improvement happens for these $s'$. Recall that we seek to optimize the long-term CVaR, where the "optimal" $s$ is actually the $\alpha$-quantile of the long-term return. We therefore propose the following operator,

$$\forall x, a, \quad \widetilde{\mathcal{T}}_\alpha U(x, a) \overset{D}{:=} \widetilde{\mathcal{T}}_{q_\alpha(U(x, a))} U(x, a).$$

This can be easily implemented through distributional RL, which we describe in the next section.

## 3 ALGORITHM

Our proposed algorithm is based on distributional Q-learning using quantile regression (Dabney et al., 2018b). It can be easily adapted to any other variants of distributional RL. Algorithm 1 shows the main algorithm for computing the loss over a mini-batch containing $m$ transition samples. Here, each distribution $\theta(x, a)$ is represented by $N$ quantiles $\theta = (\theta_1 \ldots \theta_N)$, each corresponds to a quantile level $\hat{\tau}_i = \frac{i - 0.5}{N}$. The quantile function $q_\alpha(\theta)$ can therefore be easily extracted from $\theta$. The loss function is based on quantile regression, where $\rho_\tau(u) = u(\tau - \delta_{u<0})$ where $\delta_{u<0} = 1$ if $u < 0$ and 0 otherwise. The key difference from the ordinary quantile-regression distributional Q-learning is our target action selection strategy for choosing $a'_k$ (Step 1(a) and (b)). For other implementation details, we refer the reader to (Dabney et al., 2018b).

---

**Algorithm 1** Quantile Regression Distributional Q-Learning for static CVaR

Input: $\gamma, \alpha, \theta, \theta'$, mini-batch $(x_k, a_k, r_k, x'_k)$ for $k = 1 \ldots m$

    1. For each $k = 1 \ldots m$,

        (a) $s_k \leftarrow q_\alpha(\theta(x_k, a_k))$

        (b) $a'_k \leftarrow \arg\max_{a'} \zeta(\theta', \frac{s_k - r_k}{\gamma})(x'_k, a')$

        (c) $\widetilde{\mathcal{T}} \theta_j(x_k, a_k) \leftarrow r_k + \gamma \theta'_j(x'_k, a'_k)$

    2. $\mathcal{L} \leftarrow \frac{1}{m} \sum_{k=1}^m \frac{1}{N^2} \sum_{i,j} \rho_{\hat{\tau}_i}(\widetilde{\mathcal{T}} \theta_j(x_k, a_k) - \theta_i(x_k, a_k))$

    3. Output $\nabla_\theta \mathcal{L}$.

---

The execution of a policy defined by $\theta$ requires an additional state information $s$, which summarizes the rewards collected so far. This is not part of the MDP state $x$ and can easily be updated after

observing each new reward. At the start of a new episode, $s$ is reset. The complete algorithm for executing the policy in one episode is given in Algorithm 2.

---

**Algorithm 2** Policy execution for static CVaR for one episode

---

Input: $\gamma, \alpha, \theta$

    1. $x \leftarrow$ Initial state

    2. $a \leftarrow \arg\max_{a'} \zeta(\theta, q_\alpha(\theta(x, a')))(x, a')$

    3. $s \leftarrow q_\alpha(\theta(x, a))$

    4. While $x$ not terminal state,

        (a) Execute $a$ in $x$, observe reward $r$ and next state $x'$

        (b) $s \leftarrow \frac{s-r}{\gamma}$

        (c) $x \leftarrow x'$

        (d) $a \leftarrow \arg\max_{a'} \zeta(\theta, s)(x, a')$

---

## 4   EMPIRICAL RESULTS

We implement Algorithm 1 and 2 and represent our policies using a neural network with two hidden layers, with ReLU activation. All our experiments use Adam as the stochastic gradient optimizer with learning rate 0.0001. For each action, the output consists of $N = 100$ quantile values. The complete code to reproduce our results is included in the supplementary material.

### 4.1   SYNTHETIC DATA

We first evaluate our proposed algorithm in a simple task where we know the optimal stationary policy for any CVaR level. The MDP has 4 states $x_0, x_1, x_2, x_3$ where state $x_0$ is the initial state and $x_3$ is a terminal state. Each state has two actions $a_0$ and $a_1$. Action $a_0$ generates an immediate reward following a Gaussian $\mathcal{N}(1, 1)$ and action $a_1$ has immediate reward $\mathcal{N}(0.8, 0.4^2)$. Clearly, $a_0$ gives a better expected reward but with higher variance. Each action always moves the state from $x_i$ to $x_{i+1}$. We use $\gamma = 0.9$ for this task.

For $\alpha > 0.63$, the optimal stationary policy is to choose action $a_0$ in all states, while for $\alpha < 0.62$, the optimal stationary policy is to choose action $a_1$ in all states. We compare our proposed algorithm for static CVaR with the optimal stationary policy at various levels of CVaR. Figure 1 (left plot) shows the results.

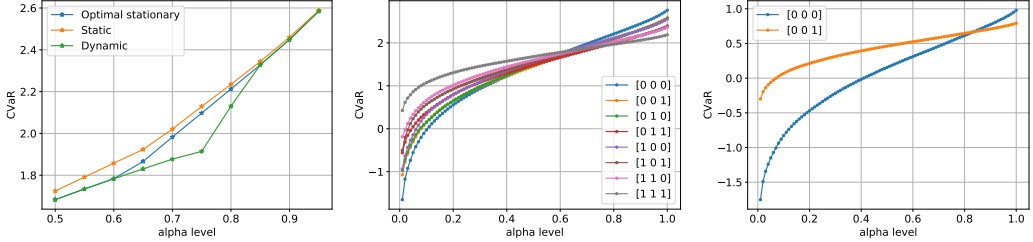

Figure 1: Left: Comparison with optimal stationary policies. Middle: Ground truth CVaR at $x_0$. Right: Ground truth CVaR at $x_2$.

We label the results for our proposed algorithm "Static" while the action-selection strategy based on (Dabney et al., 2018a) "Dynamic". We clearly see that the "Static" version outperforms "Dynamic" at all tested CVaR levels. In fact, our algorithm performs even better than the optimal stationary policy. Recall that the optimal CVaR policy may be non-stationary, where the actions in later states depend on the rewards collected so far. This example shows that learning using Algorithm 1 and execution using Algorithm 2 can indeed result in non-stationary policies by storing the "extra" information within the value distribution.

Further insights are revealed in the middle and right plots of Figure 1. These are the ground truth CVaR values for all the stationary policies, where $[1\,0\,0]$ means always choosing action $a_1$ in $x_0$ and $a_0$ in the next two states. Notice the switching point around $\alpha = 0.625$ in the middle plot and around $\alpha = 0.83$ in the right plot. The "Dynamic" CVaR action-selection strategy will choose action $a_1$ in $x_2$ for $\alpha < 0.83$ since this is the better action if one ignores the rewards collected since the beginning. However, this results in a rather conservative strategy since the optimal strategy should still favor $a_0$ in $x_2$ for $\alpha > 0.625$.

## 4.2 OPTION TRADING

We evaluate our proposed algorithm on the non-trivial real-world task of option trading, commonly used as a test domain for risk-sensitive RL (Li et al., 2009; Chow & Ghavamzadeh, 2014; Tamar et al., 2017). In particular, we tackle the task of learning an exercise policy for American options. This can be formulated as a discounted finite-horizon MDP with continuous states and two actions. The state $x_t$ includes the price of a stock at time $t$, as well as the number of steps to the maturity date, which we set to $T = 100$. The first action, "hold", will always move the state one time step forward with zero reward, while the second action, "execute", will generate an immediate reward $\max\{0, K - x_t\}$ and enter a terminal state. $K$ is the strike price. In our experiments, we use $K = 1$ and always normalize the prices such that $x_0 = 1$. At $t = T - 1$, all actions will be interpreted as "execute". We set $\gamma = 0.999$, which corresponds to a non-zero daily risk-free interest rate.

We use actual daily closing prices for the top 10 Dow components from 2005 to 2019. Prices from 2005-2015 are used for training and prices from 2016-2019 for testing. To allow training on unlimited data, we follow (Li et al., 2009) and create a stock price simulator using the geometric Brownian motion (GBM) model. The GBM model assumes that the log-ratio of prices follows a Gaussian distribution $\log \frac{x_{t+1}}{x_t} \sim \mathcal{N}(\mu - \sigma^2/2, \sigma^2)$ with parameters $\mu$ and $\sigma$, which we estimate from the real training data.

For each algorithm, each stock and each CVaR level, we trained 3 policies using different random seeds. The policies are then tested on the synthetic data (generated using the same training model) for 1000 episodes. The policies are further tested on the real data, using 100 episodes, each with 100 consecutive days of closing prices. The episode's start and end dates are evenly spread out over the 4 years of test period. All results are averaged over the 3 policies and over the 10 stocks.

Figures 2 and 3 show the test results on synthetic and real data, respectively. Again, we label the algorithms "Static" and "Dynamic" as in the previous section. Clearly, when tested on both synthetic and test data, the "Static" algorithm performs better across various CVaR level. The gap is especially significant at lower $\alpha$ levels. Also included are the results from training using $\alpha = 1$, and tested on all $\alpha$ values. This corresponds to the standard action-selection strategy based on the expected return. The learned strategies perform badly at low $\alpha$ levels, suggesting that they are taking too much risk.

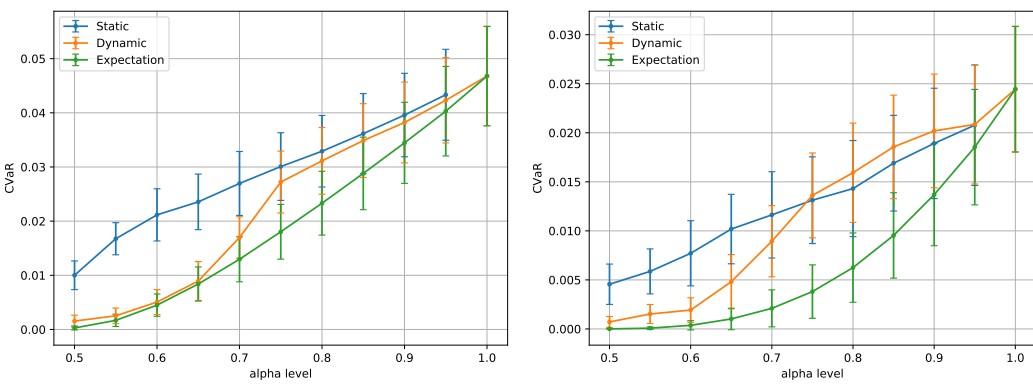

Figure 2: Test results on synthetic data          Figure 3: Test results on real data

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
