# OpenReview forum: "Distributional Reinforcement Learning for Risk-Sensitive Policies"
_ICLR.cc/2021/Conference — Reject_

### Official Review · AnonReviewer3 · 2020-10-26
**Distributional RL for optimal static CVaR policy versus markovian/dynamic CVaR**

**Rating:** 7
**Confidence:** 4

**Review:**

This paper addresses the problem of learning the optimal policy for static CVaR with distributional RL.
The authors underline the difference between static CVaR which is the CVaR on the whole cumulative discounted rewards and the dynamic CVaR which is a Markovian alternative proposed by (Ruszczynski 2010).
As pointed out by the authors, the optimal policy for static CVaR must take into account the accumulated rewards to take decisions: the optimal policy is hence not stationary.  A nice property of dynamic CVaR is that its optimal policy is stationary.
A key result of the paper is to show that optimal solution for the dynamic CVaR is suboptimal for static CVaR (Proposition 1).
Leveraging on this remark, the authors propose an algorithm based on distributional (quantile regression) to solve the static CVaR problem. The obtained policy is stationary on an augmented MDP where the states are decorated with he reward collected so-far.
The experiments on synthetic and real data underline the relevance of the proposed approach.

I really enjoyed reading this theoretical paper. It is very well written and easy to follow. The ideas presented are interesting.
My only criticisms 'or questions' are about the state augmentation:
1. As mentioned on page 4 it would be sample inefficient but one could solve the augmented MDP directly. Why not provide this approach in your experiments ?
2. As mentioned on page 5 you are using the distribution estimates to "store the information needed", instead of using state augmentation. But estimating a distribution is a costly task. Do you have any argument to justify that this approach is indeed more "sample efficient" than a naive state augmentation version without distributional estimates ? Any convergence bound ?

Minor remark:
I would be nice to have at least a small conclusion or perspective

---

> ### Author Response · Authors · 2020-11-18
> **Re: Distributional RL for optimal static CVaR policy versus markovian/dynamic CVaR**
>
> Thank you for your thoughtful comments.
>
> 1. We did start this work by trying to solve the augmented MDP directly but the results were pretty bad -- nowhere close to what we obtained here. We decided not to show these. Our code can be modified to run this as well so we welcome any interested reader to reproduce or improve on our results.
>
> 2. This is a very good question. Our view is that this is a tradeoff between computational complexity and sample complexity. It has been previously shown [Lyle et al 2019] that under tabular or linear function approximation, there is no advantage to using the entire distribution compared to just the Q-value when the objective is to optimize for the expected return. The difference, however, can be significant under non-linear function approximation and the conjecture is that by keeping track of the entire distribution (more computational work done), additional robustness is built into the solution. We believe that the phenomenon is even more prominent here in the case of CVaR objectives.

---

### Official Review · AnonReviewer4 · 2020-10-28
**Recommendation to Marginally Reject**

**Rating:** 5
**Confidence:** 3

**Review:**

This paper consider the problem of learning a risk-averse policy base on CVaR measure using distributional reinforcement learning. The main contributions of this paper are twofold. First, they show that the standard distributional RL algorithm overestimate the dynamic, Markovian CVaR, which might be too conservative. Secondly, they propose a modified algorithm that can learn a proper CVaR-optimized policy based on static, non-Markovian CVaR.

Overall this paper is well-written and easy to follow. The problem is well-motivated, and the proofs of the main propositions are clean and easy to check.

However, I have two main concerns on this paper. First, the theory part of this paper (proposition 1 & 2) are quite straight forward, and the modifications to the existing algorithm in [1] are mild, thus the novelty of this work is somewhat limited. Second, the option trading experiment train on a mixture of real stock prices and simulated stock prices (the authors use simulated data to allow training on unlimited data), and we don’t know the exact size of training set, this seems a bit wired to me. Is there any sampling complexity guarantee for the proposed algorithm? How does the performance of the algorithm scale with the training sample size？

[1]  Will Dabney, Mark Rowland, Marc G. Bellemare, and R´emi Munos. Distributional reinforcement learning with quantile regression.  In Proceedings of the Thirty-Second AAAI Conference on Artificial Intelligence.

---

> ### Author Response · Authors · 2020-11-18
> **Re: Recommendation to Marginally Reject**
>
>
> Thank you for your thoughtful comments.
>
> One of the main "selling points" in the proposed algorithm is that it only requires mild modification to the existing distributional RL algorithms. However, we do not think these modifications are trivial, and as shown by our analysis as well as the empirical results, the effect can be significant.
>
> As pointed out in Section 4.2, we use actual daily closing prices from 2005 to 2015 for training and 2016-2019 for testing. For testing on real data, these are the exact prices used. For training, instead of sampling from only 10 years of daily prices (only 2K+ data points), we fit a model to them and sample from the model instead. Further details can be obtained from the provided code, where all our results can be reproduced. Regarding sample complexity, performance improves with the training size as expected but we do not have any sampling complexity guarantee.

---

### Official Review · AnonReviewer1 · 2020-10-29
**Review of the paper**

**Rating:** 5
**Confidence:** 4

**Review:**

This paper is about risk-sensitive RL based on the CVaR risk measure. This paper is mainly based on the work presented in Dabney et al. in 2018 which is about distributional RL for a family of risk measures which includes CVaR as well. The main motivation for this work was the point that the method presented in Dabney et al. 2018 overestimates the dynamics and could be excessively conservative in certain scenarios. Authors have proposed to use static CVaR instead and have developed algorithms to do that.

This paper has solid theoretical results (propositions 1 and 2). Authors have identified a problem in Dabney et al. and proposed an algorithm to resolve it.

The major issue though, is about the evaluation and experimental results. authors have provided results on a synthetic dataset and a real dataset related to options trading. Even on the real data results, for larger values of alpha, Dabney et al. 2018 has outperformed the proposed approach. In order to make this paper ready for a venue such as ICLR, authors should provide a more comprehensive evaluation of their methods. At least, it is expected to show the performance of their approach and its comparison vs Dabney et al. on several Atari games. Otherwise, the contribution of this paper would be limited.

---

> ### Author Response · Authors · 2020-11-18
> **Re: Review of the paper**
>
> Thank you for your thoughtful comments.
>
> On the real dataset for option trading, we note that both our proposed algorithm and Dabney et al's algorithm perform very similarly for large alpha -- note the overlapping errorbars. In fact, we can tweak the results in our favor by choosing different trading periods but we choose not to do so since for large alpha we expect very similar results anyway. Our main focus is on small alphas, where the performance gap between the two is statistically significant.
>
> We agree that experimental results on larger domains would be illuminating. We indeed are working on producing more results in future work but we believe that our present results in this paper are already useful contributions to the research community.

---

### Official Review · AnonReviewer2 · 2020-10-31
**Convergance and Novelty**

**Rating:** 5
**Confidence:** 3

**Review:**

First I wanted to thank authors for putting the manuscript together, I enjoyed reading it.

Summary : Authors proposed using DRL to find a good (optimal) CVaR policy, by pointing that the optimal CVaR policy is non-stationary, and DRL can be leveraged to learn and execute this kind of policies. In addition they showed just taking a max over CVaR rather than E will not result in an optimal policy.

Strength :
1. Paper is well written, it's easy to follow and it provides necessary background for the reader to follow.
2. Important Issue : I believe finding a scalable way to optimize for CVaR is an important problem to tackle, as most of the previous work are not scalable (e.g. Chow et al)

Weakness/ Concerns:
1. Convergence: I have a concern about the convergence of the algorithm, by applying the proposed algorithm, is there any guarantee for convergence? Even in the case of policy evaluation and not control (not taking max, picking action on \pi) is there a convergence guarantee? Or if not can authors provide intuition/ reason why is that the case?

2. Novelty: Reading the paper, I challenge the novelty of current algorithm/ manuscript. Or maybe I had a hard time pinpointing it, to the best of my knowledge most of the claims have been already known, can authors please explain what they think their main contribution is? (I'm happy to change my score given the explanation)

3. Experiments: The premise of using DRL for CVaR is mainly "scalability" so that we can solve larger problems (state space mainly). However, this is not reflected in the experiments, and experiments are in small domains. I think the paper can benefit from an experiment in larger state space.

Score:
At this point I think the manuscript is not ready for a publication, however, I did enjoy reading it, and I think it's a great work so far with potentials. I am happy to change my score given authors response to my concerns.

Thanks.

---

> ### Author Response · Authors · 2020-11-18
> **Re: Convergance and Novelty**
>
> Thank you for your thoughtful comments.
>
> 1. Convergence: For the case of policy evaluation (fixed pi), convergence is guaranteed for distributional RL. This result has been proven in [Bellemere et al 2017]. In the case of optimal control (either the expected return or the CVaR), convergence in general is not guaranteed. The intuition is this: there can be more than one optimal policy (with the same expectation or CVaR) with different value distributions. Since we work in the space of value distributions, convergence would imply that we consistently favor one policy, which cannot be achieved in general but may be achievable with a consistent tie-breaking strategy -- see [Bellemere et al 2017] for more on this. This is an important issue that we intend to pursue in a future work.
>
> 2. Our main contribution is the proposed algorithm itself, which, to the best of our knowledge, is novel. The existing distributional RL algorithms allow a straightforward way to select actions by the CVaR of the value distribution, but we have shown that this leads to a solution closer to the dynamic CVaR. One naturally asks whether we can steer the solution towards optimizing the static CVaR -- we showed how, and the modification required is non-trivial.
>
> 3. We agree that experimental results on larger domains would be illuminating. We indeed are working on producing more results in future work but we believe that our present results in this paper are already useful contributions to the research community.

---

### Author Response · Authors · 2021-02-03
**Correction**

To anyone interested, we note that Proposition 1 in the paper is wrong. The proof is wrong. We will present new results in a future version of the work.

---

> ### Author Response · Authors · 2022-09-15
> **New version of the work (To appear at NeurIPS 2022)**
>
> To anyone interested, please refer to our new version instead.

---

### Decision · Program_Chairs · 2021-01-07
**Final Decision**

**Decision:**

Reject

**Comment:**

The reviewers found found the paper well motivated and well written, they found both the theoretical contributions limited in novelty and the experiments too rudimentary to be insightful.